# Detection of a Locally-Acquired Zika Virus Outbreak in Hidalgo County, Texas through Increased Antenatal Testing in a High-Risk Area

**DOI:** 10.3390/tropicalmed5030128

**Published:** 2020-08-05

**Authors:** Steven Hinojosa, Alexander Alquiza, Clarissa Guerrero, Diana Vanegas, Niko Tapangan, Narda Cano, Eduardo Olivarez

**Affiliations:** Hidalgo County Health and Human Services Department, Edinburg, TX 78542, USA; alexalquiza1@gmail.com (A.A.); Clarissa.Guerrero@hchd.org (C.G.); Diana.Vanegas@hchd.org (D.V.); Niko.Tapangan@hchd.org (N.T.); Narda.Cano@hchd.org (N.C.); Eddie.Olivarez@hchd.org (E.O.)

**Keywords:** Zika, tropical disease, epidemiology, border health, outbreak

## Abstract

Hidalgo County (HC), located along the Texas–Mexico border, was listed as a high-risk county for Zika virus (ZIKV) in 2017 by the Texas Department of State Health Services, based on its historical presence of Dengue. Due to its subtropical climate, active binational travel, and population of low socioeconomic status, Hidalgo County focused on disease detection activities for the prevention of further transmission. Therefore, Hidalgo County Health and Human Services enacted public health surveillance, reviewed laboratory results, and conducted epidemiological investigations from 2016 to 2018. In 2017, Hidalgo County experienced a locally-acquired outbreak of Zika virus disease, resulting in the highest local mosquito-borne acquisition case count for the year within the United States. This resulted in Hidalgo County reviewing epidemiological data for disease detection and risk areas. With the data review, key outcomes of testing were identified. This included the importance of both RT-PCR and IgM-ELISA/PRNT testing methods. In addition, increased antenatal testing and surveillance also recognized the need of improved disease identification and testing among the general population, especially during localized outbreaks.

## 1. Introduction

Hidalgo County fosters a transient, unique community supplemented by common travel along the Texas–Mexico border and multiple demographic factors that culminate in a high-risk population for arboviral diseases [1]. The majority of the Hidalgo County population is Hispanic or Latin at 91.8% (770,794) [2,3]. The median age is 28.9 and it has the 5th largest adolescent population in Texas. A pronounced proportion of the population was in poverty at 31.8%, more than double the national average (14.6%) and nearly double the average of Texas (16.0%). Furthermore, the median household income was well below both the state and national median by about USD 20,000 per year. Also, a great part (31.6%) of the civilian noninstitutionalized population of Hidalgo County lived with no health insurance, triple the rate of the country and nearly double the rate of the state. In addition, there was an educational disparity in Hidalgo County with only 63.7% of the population (25 and older) possessing a high school degree or higher; the state’s and country’s numbers are well above 80%. Over half of the population was female (51.1%) with about 42.2% of them between ages 15 and 44 [2,3]. The birth rate in Hidalgo County of women in this age range is 5.78% whereas the state and national averages are 5.65% and 5.21%, respectively. Lastly, Hidalgo County fosters a tropical environment suitable for the presence of the mosquito species *Aedes aegypti* and *Aedes albopictus*, the primary vectors of the Zika virus [4,5].

In April 2017, the Texas Department of State Health Services (DSHS) released a health alert, urging increased testing of pregnant women and symptomatic individuals in high-risk counties, resulting in a large spike in Zika virus testing. Recommendations were set forth by the Texas DSHS for residents of counties within the Lower Rio Grande Valley (i.e., Cameron, Hidalgo, Starr, Webb, Willacy, and Zapata) [6]. This included testing of pregnant women in their first and second trimester as a part of their prenatal care, and of symptomatic individuals presenting with a rash and at least one other common Zika virus (ZIKV) symptom, either fever, joint pain, or conjunctivitis (red eye) [5]. In April 2018, this health alert was updated to include additional high-risk counties in addition to the previous counties (i.e., Kinney, Maverick, and Val Verde counties). In this 2018 update, asymptomatic pregnant women were recommended they be tested three times during their pregnancy, once per trimester using RT-PCR only instead of paired RT-PCR and IgM-ELISA samples as recommended in 2017. The general population testing recommendation stayed the same in regards to symptom-based testing, encompassing rash and one other symptom associated with ZIKV (i.e., joint pain, conjunctivitis, fever) [7].

Hidalgo County is comprised of a younger population (over 5 years below both the state and national medians) with education and income disparities in addition to frequent cross-border travel and inherent environmental risks. Hidalgo County has also had historical evidence of Dengue transmission, another arboviral illness transmitted by the mosquito species *Aedes aegypti* and *Aedes albopictus* [8,9]. Thus, the Hidalgo County population presents as a vulnerable population with concerns of high-risk Zika virus transmission. Due to these risk factors within this population, local testing efforts for ZIKV increased and an outbreak of local acquisition was identified in the later part of 2017. This resulted in Hidalgo County having the highest local mosquito-borne acquisition case count, with four of the total seven symptomatic disease cases within the United States for 2017 [10]. This surveillance report aimed at pinpointing Zika virus cases, understanding routes of transmission, and analyzing the representation of different sources and types of testing. With this information, the goal was to establish and quantify the most effective testing measures to prevent spread of the virus. From there, implications on future surveillance and policies would better serve to protect the community in a preventative capacity to stop any locally-acquired outbreaks from occurring. High-yield tests and testing sources would be emphasized to create maximum outreach. Additionally, the data obtained and conclusions formed may assist in further scientific research conducted on similar communities.

## 2. Materials and Methods

A surveillance report was conducted to assess ZIKV testing outcomes of patients in Hidalgo County, Texas. The work conducted was considered non-research Public Health Surveillance; therefore, it was not subject to IRB review requirements. Electronic lab results were submitted to the National Electronic Disease Surveillance System (NEDSS, v5.4.6-GA, Centers for Disease Control and Prevention, Atlanta, GA, USA) by laboratories performing Zika virus tests on serum and urine samples. Commercial laboratories automatically sent positive and presumptive positive IgM-ELISA specimens to Centers of Disease Control and Prevention (CDC) laboratories for confirmation by plaque-reducing neutralization testing (PRNT). This PRNT method would assess antibodies of Dengue-1, Dengue-2, and ZIKV to assess possible cross-reactivity. CDC laboratories would then submit these laboratory results to the NEDSS database. Hidalgo County Zika virus staff ran and exported a report to abstract data from 1 January 2016 to 31 December 2018, reflecting all electronic ZIKV lab reports for this time period.

All cases were defined as either positive or negative. Then, each was designated a test type as IgM-ELISA, RT-PCR, or PRNT. Furthermore, the data were categorized into specimen type as serum, urine, or other (including umbilical blood and seminal fluid) as well as the facility type as hospital, obstetrics and gynecology (OB-GYN), unknown OB-GYN, private clinic, and public health.

Microsoft Excel (v2016, Microsoft Corp., Redmond, WA, USA) and StataIC (v15.1, 64-bit, StataCorp LLC, College Station, TX, USA) programs were used to analyze the extracted data from NEDSS. These data were evaluated with descriptive statistics on the total number of cases, median age, age range, percentage of female patients, percentage of patients tested by OB/GYNs, travel history, and the change in tests ordered in 2017 and 2018 after a Texas DSHS Zika virus Health Advisory updated testing recommendations for pregnant women to once per trimester. Texas DSHS travel history was defined as no travel history within the last 12 months since date of collection. Descriptive statistics were also used to identify the different case characteristics from the 2017 Zika virus outbreak.

## 3. Results

Testing for ZIKV among patients in Hidalgo County began in June 2016. During this time period, until 31 December 2018, a total number of 29,045 lab results came from a total of 15,016 patients. Some patients received multiple lab tests in compliance with DSHS recommendations to test during each trimester. The median age of these patients was 27 years old (*n* = 29,014), excluding 31 cases where date of birth was not collected. The age range of these lab results ranged from birth (0 days) to 99 years old. The majority of specimens (55.5%) were RT-PCR testing, as noted in Table 1. The vast majority (98.31%) of patients tested were female patients (*n* = 28,569), shown in Table 2. With new testing recommendations and capacity, testing increased 54.3 times more in 2017 compared to 2016.

In 2017, a total of 14,614 specimens were processed for residents in Hidalgo County. Figure 1 shows the 2017 trends in testing outcomes by month, where an increase in ZIKV testing and non-negative test results were witnessed, beginning April 2017, after the Texas DSHS Health Advisory. Of the specimens tested, 110 of the 117 presumptive positive laboratory specimens were forwarded to the CDC for confirmation. As per Texas DSHS epidemiological case criteria, CDC confirmation was needed to check for false positives and cross-reactivity. Of these forwarded presumptive results, 22 samples were confirmed as positive by PRNT, while all other of these IgM sample results were identified as false-positive. This resulted in 16 unique and unduplicated ZIKV patients. An additional two RT-PCR cases were also positive, but they were not subject to CDC confirmation due to their confirmatory status. These cases were divided into two categories, as defined by Texas DSHS, Zika disease cases and Zika infection cases. Zika disease cases were defined as having clinical symptomatic evidence of the virus along with laboratory confirmation. Zika virus infection cases were defined as having laboratory confirmation without clinical symptomatic evidence. In Table 2, the breakdown of case type (symptom presence), acquisition, and pregnancy status is described. Of the 18 cases, eight were symptomatic disease and ten were asymptomatic infection; eleven of these cases were identified as travel-associated (61.11%). Furthermore, most of the cases involved a pregnant patient (72.22%). Only five of the confirmed cases were identified as locally-acquired, where no international travel history was identified within the last 12 months, which indicates potential transmission within the Hidalgo community. Only two cases’ origins remained undetermined, as shown in Table 3. The five locally-acquired cases occurred in four different cities, which included McAllen, Pharr, Alamo, and Mercedes, Texas. These communities did not present similar geographical or demographical findings, presenting an outbreak within multiple local communities, rather than one specific neighborhood.

Of these 18 cases, ages ranged from 6 to 56 years, with an average age of 24.2 years and a median of 22.0 years. When assessing gender, 2 cases were male and 16 were female. In regard to travel, seven cases had travel history to neighboring Mexico cities in Tamaulipas, three cases to Nuevo Leon, and one case to both southern Mexico and Honduras. Of these 18 cases, 16 were identified as serum IgM-ELISA presumptive positive, along with CDC PRNT positive results, and 2 samples were positive through urine and serum via RT-PCR.

In 2018, testing stayed consistent with 14,155 specimens. RT-PCR testing slightly increased, while IgM-ELISA testing slightly decreased, as shown in Table 1. This included a total of nine Zika virus cases. One case was symptomatic disease and eight were asymptomatic infections. Three cases were identified as local acquisition, one case with unknown acquisition, and five cases identified as travel-associated. All cases were pregnant at the time of testing. However, in 2019, testing began to show a decrease with a total of 9323 results, with three Zika virus cases identified. All three cases were asymptomatic infection and were pregnant. One case had unknown acquisition and the other two cases were identified as travel-associated.

## 4. Discussion

Increased efforts of testing, through the Texas Zika virus public health alert and provider education, assisted in the identification of local acquisition ZIKV cases within Hidalgo County, Texas. Prior to this health advisory, asymptomatic testing for Zika virus was not routine. Therefore, with this advisory and new testing recommendations, asymptomatic Zika infections, which may not have originally been identified, were detected. In addition, with 75% of Zika disease cases and 70% of Zika asymptomatic infection cases being pregnant at the time of testing, this poses a question as to what could be the true case representation of ZIKV cases in the non-pregnant population. Therefore, increasing both surveillance efforts and disease awareness are necessary for the medical community, especially at times of outbreaks, in regards to testing and identification of cases within the general population. In addition, the majority of the cases (16 of the 18) were serology-based testing, accompanied by PRNT confirmation. Therefore, exposure time periods may be less clear. However, with multiple infections occurring in the later part of 2017, exposure may have taken place around this time. Serology-based testing produced a higher rate of false-positives after CDC testing confirmed these samples to be actually negative. However, although RT-PCR testing is the recommended method, without these serology-based testing methods, these cases may have never been identified through routine provider testing and public health surveillance. Therefore, utilization of serology, paired with PRNT testing to account for Dengue cross-reactivity, may be beneficial in future continued surveillance efforts, especially during times of increased arboviral activity within the community, compared to PCR alone. This testing method is helpful during times of Dengue outbreaks, where outbreaks of local acquisition were identified in Hidalgo County in 2013 and 2019. Furthermore, OB/GYN providers testing partners of pregnant females provided a strong surveillance system for identification of infection, and assisted in preventative education for negative partners. In addition, new testing recommendations also assisted in the detection of an asymptotic male partner of a pregnant female. Although this outbreak was limited in the number of cases with local acquisition, these findings present the importance of continued surveillance efforts along the southern Texas–Mexico border for Zika virus and other mosquito-borne illnesses.

## Figures and Tables

**Figure 1 tropicalmed-05-00128-f001:**
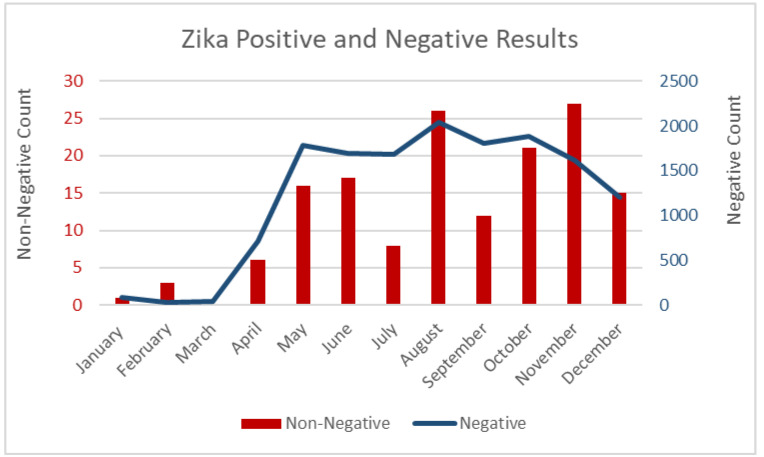
The line depicts a sharp increase in the number of Zika virus (ZIKV) specimens tested (noted as a blue line) and non-negative results (noted as red bars), after the 2017 Zika virus Health Alert was released. A total of 14,614 labs were processed, resulting in 150 non-negative samples. Non-negative samples included results that were positive, equivocal, indeterminate, or other results that were not negative. Testing peaked in August 2017, with 26 non-negative samples and 2036 negative samples.

**Table 1 tropicalmed-05-00128-t001:** Zika virus testing by test type, 2016–2018. This table depicts testing by test type and total positives by test type. Plaque-reducing neutralization testing (PRNT) took place for IgM non-negative specimens. IgM samples had a 2.18% positivity rate, while RT-PCR samples had a 0.11% positivity rate. Overall positivity rate for these specimens was 1.16%.

	IgM-ELISA	RT-PCR	PRNT	Total Results
2016	70 (25.36%)	196 (71.01%)	10 (3.62%)	276
2017	6649 (45.50%)	7855 (53.75%)	110 (0.75%)	14,614
2018	6015 (42.49%)	8026 (56.70%)	114 (0.81%)	14,155
Total Tested	12,734 (43.84%)	16,077 (55.35%)	234 (0.81%)	29,045
Total Positives	277 (2.18%)	17 (0.11%)	44 (18.80%)	338 (1.16%)

**Table 2 tropicalmed-05-00128-t002:** Gender representation of Zika virus specimen testing by facility type from 2016 to 2018. Testing with OB-GYN clinics was identified as the most common facility type for testing both females and males. Males had the lowest testing count at private laboratories (without a facility) and private clinics. Males represented only 1.69% of the specimens tested.

Source of Testing (2016–2018)						
	OB-GYN	Hospital	Lab, no Facility	Private Clinic	Public Health	Total
Female	23,807	1545	43	1694	1465	28,554 (98.31%)
Male	225	100	8	58	100	491 (1.69%)
Total	24,032 (82.74%)	1645 (5.67%)	51 (0.18%)	1752 (6.03%)	1565 (5.39%)	29,045

**Table 3 tropicalmed-05-00128-t003:** This table presents the breakdown of cases that were symptomatic disease cases versus asymptomatic infections. Acquisition was also determined, with 5 cases identified as locally- acquired. The majority of cases were pregnant with 75% pregnant symptomatic disease cases and 70% asymptomatic infection cases.

2017 Zika Case Count	Travel-Associated	Locally-Acquired	Acquisition Undetermined	Total	Pregnant
Zika Disease Cases	4	4	0	8	6 out of 8
Zika Infection Cases	7	1	2	10	7 out of 10

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
