# Peer review of "Detection of a Locally-Acquired Zika Virus Outbreak in Hidalgo County, Texas through Increased Antenatal Testing in a High-Risk Area"

_tropicalmed, 2020, doi:10.3390/tropicalmed5030128_

Round 1

Reviewer 1 Report

I feel that the authors did a good job of incorporating the suggestions of the reviewers, resulting in a substantially improved manuscript.  My one final quibble is the the legends for tables 1 and 2 need to be expanded to be more descriptive. For example, in table 1 is the number the number of tests or the number of positives.  You could say total tests (% positives) or similar text.  Tables should always be able to stand alone. Also, label the 2 y axis or differentiate by color. 

Author Response

Reviewer 1

My one final quibble is the legends for tables 1 and 2 need to be expanded to be more descriptive. For example, in table 1 is the number the number of tests or the number of positives. You could say total tests (% positives) or similar text. Tables should always be able to stand alone. Also, label the 2 y axis or differentiate by color.

Response: Figure 1 updated to have distinction between the two Y axes. Tables and figures also expanded in their description.

Reviewer 2 Report

Hinojosa et al. have compiled a report on ZIKV testing carried out between 2016 and 2018 in a high risk county in Texas, USA. The authors report on the number of diagnostic tests (RT-PCR, IgM and PRNT) carried out and the number of positives in addition to the source of testing. These data  highlight locally acquired infections and provide some insight into the source and demographics of positive tests. The manuscript should provide clarity surrounding the methods and could be improved by inclusion of some discussion about relative merits/accuracy of the various diagnostic tests.

General comments:

The study period is 2016-2018. Table 1 and 2 summarise the 3 year period. Then figure 1 and 2 and almost the whole results section only mentions 2017. Since cases were still being reported in December of 2017 and the whole period from 2016-2018 was analysed could 2018 results be incorporated? Particularly as there were additional positives according to Table 1.

Some of the numbers are confusing or appear erroneous (see specific comments below)

Specific comments:

Line 14: replace ‘conducting’ with ‘conducted’

Line 18: replace ‘this’ with ‘the’

Lines 25-40 (paragraph 1) I think most of the information regarding the population could be better summarised in a table

Line 29: multiple uses of ‘of’ after ‘double…’ or ‘triple…’ should be removed (line 30+33 too)

Line 54: Younger with regards to who? The state overall?

Line 79-80: this sentence is a little unclear/vague.

Line 93: replace ‘…labs resulted…’ with ‘…lab results…’?

Line 93 it’s not clear why only 28,973 tests are included in Table 1 when 29,063 were done.

Line 99: Authors should mention 2018 before referring to 2019, which is outside the study scope.

Line 100: began to show… a total of 9,323…

Line 107: 14,716 samples processed in 2017 but table 1 states 14,553 – why are these values different?

Lines 107+124: the methods section suggests only IgM+ were sent to CDC for PRNT testing - Is this correct? However, line 124 states that only 16/18 of the CDC confirmed positives were IgM/PRNT+. With the other 2 being RT-PCR positive. Why were the 2 RT-PCR+ samples sent to the CDC and how were they confirmed by the CDC?

Line 108: 120 samples sent to CDC for PRNT in 2017 but table 1 states 105 – why are these values different?

Line 108: ‘...potential laboratory specimens…’ = ‘…potential positive laboratory specimens…’?

Line 108: 18/120 potential positives (by IgM?) were confirmed by PRNT (15%) – Should we expect 85% of other IgM+ to be false-positive and not indicative of ZIKV infection? Or were these 120 samples low IgM+ and therefore more likely to be false-positives?

Line 115: replace ‘with’ with ‘within’

Line 121-122: ‘median’ written twice

Line 142: ‘this presents enquiry’ should be reworded.

Table 1: 53/222 PRNT tests were positive but the percentage is calculated as 4.79% (should be 23.87%)?

Table 1 (cont.): The methods state that commercial labs send IgM positive samples (or presumptive positive) to CDC for PRNT, is this the case with all labs? Why are so many tested by IgM but so few with PRNT? Are IgM+ ever considered as genuine ZIKV infection in the absence of PRNT? How do the CDC get around the problem of cross-reactivity to other flaviviruses and how accurate are the tests in this situation?

Author Response

Reviewer 2

The manuscript should provide clarity surrounding the methods and could be improved by inclusion of some discussion about relative merits/accuracy of the various diagnostic tests.

Response: Discussion expanded in Lines 168-169 to include the outcomes of testing methods, and the balance between sensitivity and case identification.

The study period is 2016-2018. Table 1 and 2 summarise the 3 year period. Then figure 1 and 2 and almost the whole results section only mentions 2017. Since cases were still being reported in December of 2017 and the whole period from 2016-2018 was analysed could 2018 results be incorporated? Particularly as there were additional positives according to Table 1.

Response: Results section added in Lines 155-162 to include 2018 and 2019 finding, and were compared to previous years.

Some of the numbers are confusing or appear erroneous (see specific comments below)

Response: Comments on the numbers presented were addressed as described in the specific comments below.

Line 14: replace ‘conducting’ with ‘conducted’

Response: The word “conducting” was changed to past tense.

Line 18: replace ‘this’ with ‘the’

Response: As suggested, the word “this” was replaced with “the”.

Lines 25-40 (paragraph 1) I think most of the information regarding the population could be better summarised in a table     

Response: Paragraph 1 kept as narrative format to describe the prevailing factors that could influence the local community’s risk for poorer health access and arboviral illness. Information also condensed to address concern of excessive wording.

Line 29: multiple uses of ‘of’ after ‘double…’ or ‘triple…’ should be removed (line 30+33 too)

Response: Instances of the word “of” before “double” and “triple” were removed to correct the error.

Line 54: Younger with regards to who? The state overall?

Response: The population of Hidalgo County is younger in comparison to both the state and national medians. The state and national medians were removed after the first set of reviewers’ comments. To clarify this point, reference was made to the state and nation without inclusion of specific numbers.

Line 79 80: this sentence is a little unclear/vague

Response: This sentence was restructured and clarified to better present the data abstraction process.

Line 93: replace ‘…labs resulted…’ with ‘…lab results…’?

Response: The sentence was updated to better reflect that a total of 29,045 resulted came from a sum of only 15,016 patients. This was meant to show that some patients were tested multiple times.

Line 93 it’s not clear why only 28,973 tests are included in Table 1 when 29,063 were done.

Response: Sentence added in lines 100-101 to present that 90 samples had test type missing.

Line 99: Authors should mention 2018 before referring to 2019, which is outside the study scope.

Response: 2018 and 2019 data were expanded to show disease and testing trends after outbreak, and kept in chronological order.

Line 100: began to show… a total of 9,323…

Response: The grammatical error was resolved with the addition of “to” and “of”.

Line 107: 14,716 samples processed in 2017 but table 1 states 14,553 – why are these values different?

Response: Data was cleaned to account for missing dates and other criteria.

Lines 107+124: the methods section suggests only IgM+ were sent to CDC for PRNT testing - Is this correct? However, line 124 states that only 16/18 of the CDC confirmed positives were IgM/PRNT+. With the other 2 being RT-PCR positive. Why were the 2 RT-PCR+ samples sent to the CDC and how were they confirmed by the CDC?

Response: The correction was made and the PCR+ and IgM+ samples were separated. A clarification was made that PCR samples were not sent to CDC for confirmation.

Line 108: 120 samples sent to CDC for PRNT in 2017 but table 1 states 105 – why are these values different?

Response: Data clean-up has taken place, and samples were corrected to reflect accurate information.

Line 108: ‘...potential laboratory specimens…’ = ‘…potential positive laboratory specimens…’?

Response: Yes, the lab specimens mentioned in this sentence were potential positives laboratory specimens. This was made clear through the addition of the word “positive”.

Line 108: 18/120 potential positives (by IgM?) were confirmed by PRNT (15%) – Should we expect 85% of other IgM+ to be false-positive and not indicative of ZIKV infection? Or were these 120 samples low IgM+ and therefore more likely to be false-positives?

Response: Information added to include description of false positives.

Line 115: replace ‘with’ with ‘within’

Response: The associated sentence was changed to clarify its purpose. 

Line 121-122: ‘median’ written twice

Response: The second instance of “median” was deleted.

Line 142: ‘this presents enquiry’ should be reworded.

Response: The aforementioned sentence was reworded for clarity.

Table 1: 53/222 PRNT tests were positive but the percentage is calculated as 4.79% (should be 23.87%)?

Response: This was an incorrect calculation. The correct percentage (23.87%) was added.

Table 1 (cont.): The methods state that commercial labs send IgM positive samples (or presumptive positive) to CDC for PRNT, is this the case with all labs? Why are so many tested by IgM but so few with PRNT? Are IgM+ ever considered as genuine ZIKV infection in the absence of PRNT? How do the CDC get around the problem of cross-reactivity to other flaviviruses and how accurate are the tests in this situation?

Response: Information expanded in Lines 117-130 in regards to testing process and procedures for presumptive IgM results. Clarification also made on case criteria of IgM results, and of cross-reactivity.

Reviewer 3 Report

General:

The authors state “This study aimed at pinpointing Zika virus cases, understanding routes of transmission, and analyzing the representation of different sources and types of testing.” And “With this data review, key outcomes of testing were identified.” The outcomes however, are only vaguely stated but should be made very clear.

Refer to figures and tables more in the text, so it is easier for the reader to look at the data.

Should introduce the abbreviation ZIKV and use it, since you are often only writing Zika instead of ZIKV, since you are talking about the virus., e.g. Zika symptoms Line 52 or Zika testing Line 57 or 69

Rephrase IgM as “IgM-ELISA” or describe the specific technique was used to assess IgM (Line 81, table 1)

Figure 2: this is a table and not a figure

Line 82 Please add a list of samples by type and the findings positivity and cross reactivity for the 18 CDC-confirmed samples

Line 108: “Potential laboratory specimens” – weird phrase, I guess what is meant is “potential ZIKV+ samples”? Were these pre-tested? If so it’s surprising that CDC only confirmed 18 of these 120 samples, please comment

Line 109: Differentiating into “Zika disease” and “Zika infection” is odd as those with disease are obviously also infected. Would rephrase “Symptomatic/asymptomatic”

Figure 1: This figure is never mentioned in the text. Why is the number 14,716 different from table 1? There is no axis label for either of the axes. And what is “non-negative”? Does this include borderline cases or why is it not called “positive”? Of those non-negative? Only 18 have been confirmed by the CDC?

Minor:

The authors mention in the beginning that Hidalgo is known for Dengue. What is the current situation, are there also ongoing transmissions of Dengue or any other Flavivirus there? Okay if beyond the scope, but would be interesting to mention.

Title: would add the word “Virus”, so it should be “Zika virus outbreak”

Line 9: you write “high risk county” – but for what? For ZIKV infection? Then you should add this here to make it clear.

Line 18: typo: this should be a the “This included THE importance”

Line 27: you start with reference 9 – why? Normally one would start with 1.

Line 31: $20,000 per year

Line 83, introduce OB-GYN abbreviation

Line 100: typo: testing began TO show a decrease. And remove comma

Table 2: Typo: Zika not zika

Author Response

Reviewer 3

The authors state “This study aimed at pinpointing Zika virus cases, understanding routes of transmission, and analyzing the representation of different sources and types of testing.” And “With this data review, key outcomes of testing were identified.” The outcomes however, are only vaguely stated but should be made very clear.

Response: Information expanded in Lines 117-130 in regards to testing process and procedures for presumptive IgM results.

Refer to figures and tables more in the text, so it is easier for the reader to look at the data.

Response: Figures and tables incorporated and referenced more within text.

Should introduce the abbreviation ZIKV and use it, since you are often only writing Zika instead of ZIKV, since you are talking about the virus., e.g. Zika symptoms Line 52 or Zika testing Line 57 or 69

Response: The abbreviation ZIKV was introduced early in the Introduction and replaced later instances of “Zika” when appropriate.

Rephrase IgM as “IgM-ELISA” or describe the specific technique was used to assess IgM (Line 81, table 1)

Response: All instances of “IgM” replaced with “IgM-ELISA” as it was the specific technique used in assessment.

Figure 2: this is a table and not a figure

Response: The label was changed from “Figure 2” to “Table 3”, and text in results updated.

Line 82 Please add a list of samples by type and the findings positivity and cross reactivity for the 18 CDC-confirmed samples

Response: Clarification was made to describe the outcomes of the CDC-confirmed samples.

Line 108: “Potential laboratory specimens” – weird phrase, I guess what is meant is “potential ZIKV+ samples”? Were these pre-tested? If so it’s surprising that CDC only confirmed 18 of these 120 samples, please comment

Response: Yes, “potential” was changed to “presumptive”. Explanation was added (Line 114) to better define private testing and CDC confirmation process.

Line 109: Differentiating into “Zika disease” and “Zika infection” is odd as those with disease are obviously also infected. Would rephrase “Symptomatic/asymptomatic”

Response: Definitions stay in current nomenclature. Line 117 updated to reflected that these are Texas DSHS defined terms. 

Figure 1: This figure is never mentioned in the text. Why is the number 14,716 different from table 1? There is no axis label for either of the axes. And what is “non-negative”? Does this include borderline cases or why is it not called “positive”? Of those non-negative? Only 18 have been confirmed by the CDC?

Response: Figure now mentioned in Line 113. Data was cleaned to account for missing dates and other criteria. Non-negative now defined in figure 1 description. Figure description also updated to denote that count represent specimens tested, and that specimens are tested multiple times for confirmatory status.

The authors mention in the beginning that Hidalgo is known for Dengue. What is the current situation, are there also ongoing transmissions of Dengue or any other Flavivirus there? Okay if beyond the scope, but would be interesting to mention.

Response: Line 157 added in context in regards to current and previous Dengue activity.

Title: would add the word “Virus”, so it should be “Zika virus outbreak”

Response: The word “Virus” was added to the title to improve clarity.

Line 9: you write “high risk county” – but for what? For ZIKV infection? Then you should add this here to make it clear.

Response: Yes, Hidalgo County was listed as a high-risk county for ZIKV. This was clarified in the appropriate sentence.

Line 18: typo: this should be a the “This included THE importance”

Response: The word “this” was replaced to “the”.

Line 27: you start with reference 9 – why? Normally one would start with 1.

Response: Reference corrected to chronological instead of alphabetical.

Line 31: $20,000 per year

Response: The phrase “per year” was added accordingly.

Line 83, introduce OB-GYN abbreviation

Response: The full title and abbreviation for OB-GYN was added.

Line 100: typo: testing began TO show a decrease. And remove comma

Response: The grammatical error was corrected as stated.

Table 2: Typo: Zika not zika

Response: The word “zika” was capitalized.

Round 2

Reviewer 2 Report

The authors have addressed the issues raised previously. Just a few minor comments:

Line 33: 'Triple the rate of the country and nearly double the rate of the state'

Line 51: “compared to paired RT-PCR and IgM-ELISA samples”. Slightly unclear why this comparison is made - was RT-PCR and IgM-ELISA the recommendation for 2017?

Line 77: ZIKV

Line 81: Still unclear what "...filtered through using different confining parameters..." specifically means? Does this refer to the sentences that follow I.e. positive or negative, test type etc.? If so, I think it should be removed as it isn't clear (at least to me). If it's an additional step then could it be made clearer what was actually done?

Table 2: State in the table legend which years the table corresponds to (is it 2016-2018?)

Line 120: ‘…while all other of these…’

Line 121: Slightly confusing. Perhaps delete ‘that were not previously counted as cases’ as it implies that there were indeed some previously counted cases in 2017 other than the two RT-PCR+ which you go on to mention.

Line 131: Replace ‘with’ with ‘within’

Line 138-139: 12 travel locations are described for the 11 travel-related cases. Is this due to some cases travelling to multiple areas?

Author Response

Reviewer Comments and Suggestions

Line 33: 'Triple the rate of the country and nearly double the rate of the state'

Response: The suggested phrases/wording was added to the appropriate line.

Line 51: “compared to paired RT-PCR and IgM-ELISA samples”. Slightly unclear why this comparison is made - was RT-PCR and IgM-ELISA the recommendation for 2017?

Response: Yes, in 2017, the recommendation by Texas DSHS was to test with both RT-PCR and IgM-ELISA concurrently. The sentence was re-written for clarification.

Line 77: ZIKV

Response: The Zika virus abbreviation was correctly changed to ZIKV.

Line 81: Still unclear what "...filtered through using different confining parameters..." specifically means? Does this refer to the sentences that follow I.e. positive or negative, test type etc.? If so, I think it should be removed as it isn't clear (at least to me). If it's an additional step then could it be made clearer what was actually done?

Response: Yes, by the mentioned quote, we were referring to the following sentences. To improve the clarity of the paragraph, the mentioned sentence was removed.

Table 2: State in the table legend which years the table corresponds to (is it 2016-2018?)

Response: Yes, the table corresponds to the years 2016-18. This was added in the table legend and title.

Line 120: ‘…while all other of these…’

Response: The word “of” was inserted into the appropriate sentence.

Line 121: Slightly confusing. Perhaps delete ‘that were not previously counted as cases’ as it implies that there were indeed some previously counted cases in 2017 other than the two RT-PCR+ which you go on to mention.

Response: For clarity, the mentioned phrase was removed as its implication was incorrect.

Line 131: Replace ‘with’ with ‘within’

Response: As suggested, the word “with” was replaced with “within.”

Line 138-139: 12 travel locations are described for the 11 travel-related cases. Is this due to some cases travelling to multiple areas?

Response: Lines 138-139 counts were corrected and clarified. Travel to southern Mexico and Honduras were the same case, and Tamaulipas and Nuevo Leon cases were also reviewed and corrected.  

This manuscript is a resubmission of an earlier submission. The following is a list of the peer review reports and author responses from that submission.

Round 1

Reviewer 1 Report

This is an interesting report showing the that increased testing of pregnant women as part of a health alert led to recognition of local transmission.

Line 28:  rewrite

Lines 30-31:  Best to say that Aedes aegypti and Aedes albopictus are the primary vectors of Zika virus.  Other species are capable of transmitting it.  Please include a reference for this statement.

Lines 41-50:  Duplicated text.

Lines 49-50:  Are these rates statistically different?

Lines 79-84:  You use the term labs rather than patients?  Are there multiple tests from the same patient included?

Lines 123-124: Although PCR testing is the recommended method, without these serology-based testing methods, these cases may have not been identified through routine public health surveillance. This is an interesting observation and should be expanded.

Clearly, most of the samples came from pregnant women due to the health alert.  This will, of course, affect the median age.  Perhaps the title should be changed to Zika Virus testing of pregnant women….

Table 1:  can you include the number of positives for each category?

Reviewer 2 Report

The manuscript tropicalmed-806121, entitled: ’Zika Virus Testing to Identify a Locally-acquired Outbreak in Hidalgo County, Texas’ describes the findings of passive surveillance for Zika virus (ZIKV) infection in Hidalgo County. The efforts resulted in detection of 18 ZIKV cases among an unknown number of persons tested (only the number of tests is reported), in a population of unknown size. The manuscript reflects a wealth of data, but it is a difficult read because of lack of structure and clarity, and questionable analysis and interpretation.

I will offer my comments in detail below:

Title: the title is promising, but it does not reflect the manuscript’s content: No locally-acquired outbreak was identified, and the methods do not permit such an analysis . The title should at least be informative on what was done in which population, and the authors could opt to confine their analysis to antenatal screening for ZIKV.

Abstract: the abstract should be concise (high risk county? For what?), and should offer the manuscript’s conclusions; highest ‘local acquisition count’ in the US? Not clear from the data presented.

Introduction: The introduction should provide background and rationale for the study. The authors commence with a description of Hidalgo county’s socioeconomic status and demographics (ie description of the population), entomological data and ZIKV testing policy (which is repeated from line 40), and pregnancy rate (or more accurate ‘birth rate’)- this data should be presented in the methods section. There are many inaccuracies (eg. Aedes spp. (in italic!) mentioned are not the ‘only ones’ capable of transmitting ZIKV, real time reverse transcription PCR (RT-PCR), rather than PCR, odd testing policy that is not in line with CDC’s or international recommendations for screening pregnant women- but may have been recommended by Texas DSHS. A change of testing policy during the study period is mentioned but not elaborated.

Materials and Methods: this is not a retrospective-observational study. General comment for this section: it should be written so another researcher could repeat the study. Which samples were sent for PRNT confirmation? What data is extracted from the NEDSS? who reviewed what? Post hoc designation of testing types? Please, clarify how data were handled and what the statistics programs were used for?

Results: How many patients were tested? The results section describes the number of tests, and we know that several were repeated, there is probably also overlap in ZIKV RT-PCR and testing for antibodies. Here, the authors introduce another method: a distinction between ZIKV disease and ZIKV infection- this distinction is difficult to assess from the case records. Where did the travel information come from? The methods used do not allow firm conclusions on the location where persons were infected. Was any of the cases exposed to ZIKV infection via sexual transmission? Please structure this section in line with what you set out to do (as in the methods section).

Discussion: Please reflect on the manuscript’s main findings. How do these fit into the existing body of evidence? What do the authors think of finding 18 cases in this population (at the expense of nearly 30,000 tests)? Any lessons for future surveillance efforts?

References: references should be used to support or reflect on your statements. Apart from census data and an entomological study, only dengue studies are referenced, and these are misplaced or absent in the manuscript.